# Multiple-Drug Resistant Shiga Toxin-Producing *E. coli* in Raw Milk of Dairy Bovine

**DOI:** 10.3390/tropicalmed9030064

**Published:** 2024-03-19

**Authors:** Safir Ullah, Saeed Ul Hassan Khan, Muhammad Jamil Khan, Baharullah Khattak, Fozia Fozia, Ijaz Ahmad, Mohammad Ahmad Wadaan, Muhammad Farooq Khan, Almohannad Baabbad, Sagar M. Goyal

**Affiliations:** 1Department of Zoology, Faculty of Biological Science, Quaid-e-Azam University, Islamabad 45320, Pakistan; saeedkhan@qau.edu.pk (S.U.H.K.);; 2Department of Microbiology, Kohat University of Science and Technology, Kohat 26000, Pakistan; dr.baharullah@kust.edu.pk; 3Department of Biochemistry, KMU Institute of Dental Sciences, Kohat 26000, Pakistan; 4Department of Chemistry, Kohat University of Sciences & Technology, Kohat 26000, Pakistan; drijaz_chem@yahoo.com; 5Zoology Department, College of Science, King Saud University, P.O. Box 2455, Riyadh 11451, Saudi Arabia; wadaan@ksu.edu.sa (M.A.W.);; 6College of Veterinary Medicine, University of Minnesota, St. Paul, MN 55455, USA; goyal001@umn.edu

**Keywords:** Antibiotic resistance, ESBL, milk, Shiga toxin-producing *Escherichia coli*, mPCR, Pakistan, virulence genes, zoonotic bacteria

## Abstract

Introduction: Raw milk may contain pathogenic microorganisms harmful to humans, e.g., multidrug-resistant *Escherichia coli* non-O157:H7, which can cause severe colitis, hemolytic uremia, and meningitis in children. No studies are available on the prevalence of Shiga toxin-producing *E. coli* (STEC O157:H7) in sick or healthy dairy animals in the Khyber Pakhtunkhwa Province of Pakistan. Aim: This study aimed to isolate, characterize, and detect antibiotic resistance in STEC non-O157:H7 from unpasteurized milk of dairy bovines in this province. Materials and Methods: We collected raw milk samples (*n* = 800) from dairy farms, street vendors, and milk shops from different parts of the Khyber Pakhtunkhwa Province. *E. coli* was isolated from these samples followed by latex agglutination tests for serotyping. The detection of STEC was conducted phenotypically and confirmed by the detection of virulence genes genotypically. An antibiogram of STEC isolates was performed against 12 antibiotics using the disc diffusion method. Results: A total of 321 (40.12%) samples were found to be positive for *E. coli* in this study. These samples were processed for the presence of four virulence genes (*Stx1*, *Stx2*, *ehxA*, *eae*). Forty samples (5.0%) were STEC-positive. Of these, 38%, 25%, 19%, and 18% were positive for *Stx1*, *Stx2*, *ehxA*, and *eae*, respectively. Genotypically, we found that 1.37% of STEC isolates produced extended-spectrum beta-lactamase (ESBL) and contained the *bla_CTX M_* gene. Resistance to various antibiotics ranged from 18% to 77%. Conclusion: This study highlights the risk of virulent and multidrug-resistant STEC non-O157:H7 in raw milk and the need for proper quality surveillance and assurance plans to mitigate the potential public health threat.

## 1. Introduction

Infections with Shiga toxin-producing *Escherichia coli* (STEC), also known as verotoxin-producing *E. coli* (VTEC), cause serious public health and economic problems globally. Food-borne *E. coli* is associated with the use of unhygienic milk, meat, and dairy products of animal origin. STEC causes diarrhea, dysentery, severe enteritis, hemorrhagic colitis (HC), and hemolytic urinary syndrome (HUS) in humans. Commensal *E. coli*, discovered in 1885, is the major facultatively anaerobic flora of the human and animal intestinal systems [1]. Diarrheagenic *E. coli* (DEC) strains can be divided into six main categories based on distinct epidemiological and clinical features, and specific virulence determinants [2]: enterohaemorrhagic *E. coli* (EHEC) or Shiga-toxin-producing *E. coli* (STEC), enterotoxigenic *E. coli* (ETEC), enteropathogenic *E. coli* (EPEC), enteroinvasive *E. coli* (EIEC),enteroaggregative *E. coli* (EAEC), and diffusely adherent *E. coli* (DAEC) [3]. STEC are a heterogeneous group of organisms characterized by the production of two potent cytotoxins, e.g., Shiga-like toxins 1 and 2 (*Stx1 and Stx2*). In some strains, the LEE locus related to the attaching and effacement lesion is also present [4].

The intimate attachment of bacteria to the host cell is mediated by the binding of intimin, the product of the (*eae*) gene, to the translocated intimin receptor. The correlation between the existence of the *Stx2* gene in the infecting strain and the occurrence of severe disease in humans has been established [5]. Experimental infection in primates has shown that the administration of purified *Stx2*, as opposed to *Stx1*, is capable of causing HUS [6]. Contact with animals frequently serves as a facilitator for the transmission of STEC from dairy farm environments to humans [7]. Hemolytic uremic syndrome (HUS) resulting from the consumption of raw milk has been reported in the United States of America (in the states of Wisconsin, Washington, and Oregon) [7], Canada [8], and Finland [9]. Despite the relatively modest percentage of raw milk consumers in Western societies (e.g., 1 to 2% in the US), the associated STEC outbreaks have been disproportionately impactful. Outbreaks attributable to *E. coli* O157-contaminated dairy products, such as a yogurt outbreak in the UK [10] and cheese outbreaks in the US (Wisconsin) and France [11], are infrequent. However, the contamination of raw milk and cheese emerges as a substantive risk to human health [12,13].

The use of antibiotics in animal production has given rise to antibacterial drug resistance, presenting a hazard to the dairy industry. Studies indicate that STEC strains isolated from raw milk and dairy products exhibit resistance to various antibiotics [14]. Thus, the emergence of pathogenic multidrug-resistant (MDR) and extended-spectrum β-lactamase (ESBL)-producing *E. coli* is a growing global concern [15]. ESBL-producing strains, encompassing variants like *bla_CTX-M_*, *bla_TEM_*, and *bla_SHV_*, curtail therapeutic options and are associated with the feces of cattle and the dairy farm environment [16]. Various studies on *E. coli* in the feces and meat of sheep, goats, and cattle have been conducted in Pakistani cities, including Islamabad, Lahore, and Peshawar [17]. These pathogens present an alarming challenge for treating generalized infections. The broad use of second- or third-generation antibiotics for bacterial infection treatment contributes substantially to antimicrobial resistance in STEC. Similarly, the increasing prevalence of extended-spectrum beta-lactamase (ESBL)-producing *E. coli* is a global concern, including in Khyber Pakhtunkhwa, Pakistan. The collection of raw milk, due to its potential for cross-contamination, elevates the risk further. Consequently, this study seeks to determine, for the first time, the prevalence of multidrug-resistant Shiga toxigenic serotypes of *E. coli* in raw milk samples collected from dairy bovines in dairy farms, milk collection centers, street vendors, and milk shops across different regions of Khyber Pakhtunkhwa.

## 2. Materials and Methods

### 2.1. Source of Samples

The sampling period was from June 2020 to January 2021. Raw milk samples (*n* = 800) were collected from dairy farms, milk shops, milk collection centers, and street vendors in various locations of the province for the isolation and identification of pathogenic STEC. Samples in 100 mL amounts were obtained in sterile screw-capped plastic bottles, placed in an ice box, properly coded based on the sample source, and then transported to the laboratory for microbiological analysis.

### 2.2. Isolation and Characterization of E. coli from Raw Milk Sample

Raw milk samples in 10 mL amounts were each mixed with 90 mL of buffered peptone water (BPW) followed by incubation at 37 °C for 24 h. After 24 h, a loopful from this culture-buffered peptone was streaked on MacConkey agar plates, which were then incubated for 24 h at 37 °C. The pinkish single colonies were further streaked on plates of eosin methylene blue agar (EMB) (Oxoide, Ltd., Hampshire, UK) plates and incubated at 37 °C for 24 h. *E. coli* growth appeared as a “metallic sheen” on EMB agar. The pre-enriched samples were serially diluted and surface plated onto Cefixime Tellurite-Sorbitol MacConkey agar (CT-SMAC) (Oxoide, Ltd., Hampshire, UK) in duplicates to detect the *E. coli* non-O157 group. The colonies with typical STEC-characteristic colorless colonies with a smoky center were transferred to nutrient agar slants and maintained for further characterization. In addition, Gram staining was carried out to confirm the identity of the *E. coli* strains. Non-sorbitol fermenting isolates were subjected to a latex agglutination test for further confirmation. The isolates were preserved in a nutrient broth with glycerol and stored at −20 °C.

### 2.3. Biochemical Characterization

Biochemical characterization was based on a series of tests including an IMVIC test (Indole, Methyl red, Voges-Proskauer (VP), and Citrate), a catalase test, and an oxidase test. The sugar fermentation tests were performed with 1% glucose, maltose, sucrose, and sorbitol as the sole carbon source.

### 2.4. Serotyping of the E. coli Isolates

Serotyping of the isolated *E. coli* strains was conducted using a rapid latex agglutination test with latex particles specific for STEC O157 (Pro-Lab Diagnostics Inc., Merseyside, UK). The result based on the agglutination of the test template within one minute was measured as a positive result as described by the manufacturer [18].

### 2.5. Phenotypic Detection of ESBL-Producing STEC

Confirmation of ESBL-producing STEC was performed using a Kirby-Bauer disc diffusion test using both Cefotaxime 30 mg and Ceftazidime disks 30 mg with and without 10 mg of clavulanate as per the Clinical Laboratory Standard Institute method. A difference of > 5 mm between the zone diameters of each disk and their clavulanate disk was calculated. Phenotypical ESBL production was confirmed using a Double Disk Synergy Test (DDST), as per published guidelines [19].

### 2.6. Detection of Shiga Toxin-Producing and ESBL Virulence Genes through Polymerase Chain Reaction (PCR)

#### Bacterial Culture

Selected *E. coli* strains were inoculated into a brain–heart infusion broth (BHI) followed by 18 h incubation at 37 °C. The broth culture of a 1mL amount was placed in sterile 1.5 mL Eppendorf tubes followed by centrifugation at 1300 rpm for 5 min. The supernatant was discarded and the pellet was re-suspended in 200 µL of nuclease-free water and then heated on a hot plate at 98 °C for 10 min followed by ice treatment. Again, the tubes were centrifuged at 13,000 rpm for 8 min. The supernatant was used as a DNA template. The purity and concentration were checked through a Nanodrop. The DNA samples were subjected to a Multiplex Polymerase Chain Reaction (mPCR) to detect the presence of *Stx1*, *Stx2*, *eae*, and *ehxA* genes [20]. Amplified DNA fragments on 2% agarose gel electrophoresis were visualized through a UV gel documentation system. The mPCR conditions are described in (Table 1).

The detection of gene markers for ESBL (*bla_CTXM_*, *bla_SHV_*, and *bla_TEM_*) was carried out using an mPCR based on the primer pairs mentioned in Table 1. The 25 µL PCR mixture was prepared by adding 2.5 µL of PCR buffer, 2.0 µL of 25 mm MgCl_2_, 0.5 µL of 10 mm dNTPs, 1 µL of Taq polymerase (3 U/µL), 17.75 µL of PCR grade water, 0.125 of respective primers, and 2 µL of the extracted DNA template. The reagents for the PCR were procured from Macrogen, Seoul, Republic of Korea. The reaction mixture was initially denatured for 10 min at 94 °C, subjected to 30 cycles of amplification at 94 °C for 1 min, annealed at 56 °C for 45 s, extended at 72 °C for 45 s, finally extended at 72 °C for 7 min, and held at 4 °C. The amplified PCR products were separated on a 2% (*w*/*v*) agarose gel by electrophoresis. Respective bands were visualized using a gel documentation unit alongside a 100 bp DNA ladder. Each of the PCR runs contained a positive control with DNA extracted from known strains of *E. coli*, obtained from the Animal Science Institute, NARC, Islamabad. The negative control consisted of 2 µL of PCR-grade water instead of the DNA template.

### 2.7. Antibiotic Susceptibility of STEC

The antimicrobial susceptibility of the isolated STEC strains was performed on Mueller- Hinton agar (MHA) (Oxoide, Ltd., Hampshire, UK) using the standard single-disk diffusion method. The inoculum of the isolated *E. coli* was prepared using normal saline, which was then adjusted to 0.05 McFarland unit of turbidity. The inoculum were spread on the MHA plates and the antibiotic discs were applied aseptically after the inoculum had dried. The antibiotic disks used were penicillin, Amoxicillin, Amoxicillin and Clavulanic acid, Cefotaxime, Cefotaxime and Clavulanic acid, Gentamicin, Streptomycin, Oxytetracyline, Sulphamethoxazole, Norfloxacin, Enrofloxacin, and Florefenical. The discs containing standard amounts of antibiotics were applied on the plates, followed by incubation at 37 °C for 24 h. The plates were then observed for the appearance of zones of inhibition, which were categorized into sensitive, (S) intermediate (I), and resistant (R). The standard breakpoints for sensitivity and resistance were adopted from the Clinical Laboratory Standard Institute guidelines [21].

### 2.8. Detection of Multidrug Resistance (MDR) among STEC Isolates

Multidrug resistance is defined as resistance to at least two of the b-Lactamase, Aminoglycoside, or quinol one antibiotics. The isolates with multidrug-resistant characteristics were ascertained by observing the resistance pattern of the isolates to the various antibiotics tested.

### 2.9. Statistical Analysis

For statistical analysis, we used the Microsoft Excel, version 10 program, employing a Student’s *t*-Test to evaluate important associations among antibiotic resistance frequencies. A *p*-value of 0.05 was considered as a statistically significant point in the analysis.

## 3. Results

### 3.1. Prevalence of E. coli in Raw Milk

Of the 800 raw milk samples tested, 321 (40.5%) were positive for *E. coli.* The conventional method of using primary enrichment on mTSB and plating showed 19.75% (*n* = 158) as NSF colonies on CT-SMAC. All non-sorbitol-fermenting STEC were confirmed using a Rapid Latex agglutination test using latex particles specific for *E. coli* O157 and non-O157:H7 as described by the manufacturer [21].

When molecular characterization through a multiplex polymerase chain reaction (mPCR) was used to test for the presence of Shiga toxin genes in the 158 non-O157:H7 isolates, the Shiga toxin genes were found in 40 of 800 (5%) raw milk samples. Among these Shiga toxin-producing *E. coli* (STEC) isolates, the distribution of virulence genes was observed as follows: *stx1* gene: 38%, *stx2* gene: 25%, *hlyA* gene: 19%, and *eae* gene: 18%. This detailed molecular analysis provides insights into the genetic composition of the identified *E. coli* isolates, indicating the presence of virulence factors associated with pathogenic strains. The region-wise prevalence ratio of STEC non-O157:H7 in raw milk samples is presented in Table 2 and Figure 1.

In terms of geographical distribution, a higher incidence of STEC non-O157:H7 was observed in raw milk samples collected from Peshawar, accounting for 13% (20/150). This was followed by Dera Ismail Khan, where the prevalence of STEC non-O157:H7 was recorded as 7% (6/85) mentioned in Table 2. The dense population of dairy animals in both Peshawar and Dera Ismail Khan may contribute to human infections. In the sampled sources, a greater occurrence of STEC non-O157:H7 was observed in raw milk samples obtained from milk shops, accounting for 6% (30/500), followed by dairy farms where the prevalence was recorded as 4% mentioned in Table 3 and Figure 2. The region-wise prevalence of STEC positive isolates is shown.

### 3.2. Multiplex PCR for Virulence Genes

A Multiplex Polymerase Chain Reaction (mPCR) revealed that isolates carried different virulence genes, (*Stx1*, *Stx2*, *eae*, and *hlyA*) with appropriate sizes of 100 bp, 150 bp, 200 bp, and 534 bp, respectively, mentioned in Table 4 and Figure 3.

### 3.3. Detection of ESBL Genes

Of the 40 STEC strains, 27.3% (*n* = 11) were positive for ESBL using the double-disc method. All phenotypically ESBL-positive STEC isolates were detected to have *bla_CTX-M_* 550 bp No *bla_TEM_* (1086 bp) or *bla_SHV_* was detected by the mPCR (Figure 4). The STEC non-O157:H7 strains producing ESBL exhibited positive results in the sampled sources, with a prevalence of 1.6% (9/500) in milk shops and a prevalence of 1% (2/200) in dairy farms. The antibiotic susceptibility test of the STEC-positive ESBL isolates showed a high resistance to ceftizoxime, ceftazidime, amoxicillin, penicillin, and Gentamicin with the range of 77–18%. The STEC-positive strains were sensitive to Enrofloxacin and Norfloxacin.

### 3.4. Antimicrobial Resistance among STEC

The 40 isolated STEC strains showed antimicrobial sensitivity as follows: Norfloxacin 59%, followed by Enrofloxacin 54%, Florefenical 50%, and the least sensitivity was recorded for Oxytetracyclinat 36%. On the other hand, the strains were highly resistant to Penicillin, Amoxicillin, Amoxicillin and Clavulanic acid, Cefotaxime, Gentamicin, Sulphamethoxazole, and Streptomycin. Most of the STEC non-O157:H7 isolates were resistant to more than three different antibiotics tested. These findings highlight the importance of continued monitoring, strict preventive measures, and the need for region-specific interventions to mitigate the risks associated with bacterial contamination in raw milk, as shown in Table 5.

## 4. Discussion

The prevalence of pathogenic STEC non-O157:H7 in raw milk is an indicator of contamination of milk and potential public health risks in the dairy value chain system. The contamination of raw milk with pathogenic *E. coli* could be from direct fecal contact, contaminated water, or other sources. This study provides critical insights into the prevalence, virulence factors, ESBL production, and antibiotic resistance of STEC in dairy milk. The overall prevalence of total *E. coli* and pathogenic *E. coli* in this study was 40.5% (*n* = 321 of 800) and 19.5% (*n* = 158) pathogenic *E. coli*, respectively, indicating that both pathogenic and non-pathogenic *E. coli* are present in raw milk. The overall prevalence of STEC non-O157:H7 was 5% (*n* = 40) suggesting that there is a high contamination of raw milk. These findings are in agreement with previous studies in Peshawar, where *E. coli* O157:H7 and *E. coli* non-O157:H7 were detected in 8.75% of branded milk samples [22].

The distribution of virulence genes by the molecular method was as follows: *stx1* gene: 38%, *stx2* gene: 25%, *hlyA* gene: 19%, and *eae* gene: 18%. This reinforces the potential severity of STEC infections. In terms of geographical distribution, a higher incidence of STEC non-O157:H7 was observed in raw milk samples collected from Peshawar (13%; or 20/150), followed by Dera Ismail Khan (7% or 6/85). This is not surprising due to the dense population of dairy animals in both Peshawar and Dera Ismail Khan.

As far as sources are concerned, a greater occurrence of STEC non-O157:H7 was observed in raw milk samples obtained from milk shops, accounting for 6% (30/500), followed by dairy farms (4% or 8/200), as shown in. A study in the city of Tandojam, Sindh also reported heavy contamination (57%) of raw milk and dairy products with *E. coli* [23]. A study that found a multidrug-resistant strain of STEC O157 in Peshawar also supports our research findings [24]. However, variations in the prevalence of *stx1* and *stx2* in different regions highlight the complex nature of STEC strains and the need for region-specific investigations [25,26]. Our study under scores a global variation in the prevalence of multidrug resistance in STEC of raw milk origins, as seen in Ethiopia [27].

Variations in the prevalence rates observed in different regions indicate the influence of diverse factors such as milk production practices, handling procedures, and hygienic conditions. The prevalence of STEC in raw milk mirrors global concerns about food-borne illnesses. STEC infections, known for causing severe gastrointestinal symptoms and life-threatening conditions like hemolytic uremic syndrome (HUS), pose a significant threat to public health [28,29].

During ESBL detection, all 40STEC non-O157:H7 isolates were subjected to confirmatory tests; 27.3% (*n* = 11) were positive for ESBL. The source of ESBL strains were milk shops at 1.6% (9/500) and dairy farms at 1% (2/200). The detection of multidrug resistance in ESBL producers is alarming [30]. This is particularly troubling given the common use of antibiotics, highlighting the potential transmission of multidrug-resistant strains through the consumption of raw milk. Because of the small number of isolates, it is difficult to establish strong epidemiological connections. The isolates, however, demonstrated the presence of the *bla_CTX-M_* gene. This finding aligns with studies reporting high levels of antibiotic resistance in dairy-related *E. coli* strains in Pakistan, further emphasizing the need for comprehensive measures to ensure food safety [31]. An increased occurrence of multidrug-resistant STEC strains in China and other cities in Pakistan has been reported. Extreme use of antibiotics for livestock management practices and conducive horizontal gene transfer systems within the *E. coli* family could be linked to the recorded high resistance. This poses a greater threat of zoonotic disease outbreaks in these countries [32].

All STEC non-O157:H7 isolates showed resistance to Amoxicillin, Cefetaxime, and Gentamicin, but were highly sensitive to Ciprofloxacin. Antimicrobial-resistant bacteria are one of the most serious public health issues and are predicted to cause the death of millions of people annually beginning in 2050 [33]. MDR strains have also been reported in China and Iran [34]. The prevalence of ESBL genes, with *bla_CTX-M_* being the most predominant, emphasizes the importance of monitoring and addressing antibiotic resistance in STEC strains to address significant challenges to food safety [35].

Raw milk has been reported to be a potential source of not only food-borne pathogens but also antimicrobial residues. The traditional system of production, handling, transportation, and marketing of milk and milk products may contribute to pathogenic multidrug-resistant *E. coli* contamination. This highlights the importance of improving hygiene practices throughout the milk production, processing, and marketing chains. Our results underscore the critical need for continued surveillance and careful use of antibiotics. The increasing prevalence of antibiotic resistance in isolates from animal origins has important therapeutic implications. Monitoring ESBL-producing enteric bacteria at various levels (animals, humans, and the environment) is crucial for the One Health approach since these bacteria not only contribute to the spread of pathogenic bacteria but also serve as vehicles for the dissemination of antibiotic resistance.

## 5. Conclusions

The findings of this study indicate the presence of multidrug-resistant STEC non-O157:H7 in raw milk sold within the Khyber Pakhtunkhwa Province. The raw milk produced, distributed, and supplied to consumers in the study area contains highly infectious and drug-resistant milk-borne pathogens, which are hazardous to public health. The major sources of pathogenic contamination in dairy bovine raw milk may be from poor sanitation practices during milking, milk collection, processing, storage, and transportation. This study also characterized various virulence gene profiles of STEC non-O157:H7 pathogenic strains isolated from raw milk. The results also indicated that these isolates were resistant to most antimicrobial drugs, which may worsen the infections caused by these pathogens. The higher prevalence of multidrug-resistant *E. coli* non-O157:H7 isolates in dairy bovine raw milk is alarming in terms of risks to public health, animal health, and food safety. In summary, the milk process from production at the farm level to the consumer level needs hygienic practices. In addition, the rational utilization of antimicrobials needs to be practiced rather than their indiscriminate use.

Based on the above situation, the following recommendations are formulated.

There is a need to improve the quality and safety of milk through hygienic practices during the handling of raw milk. Raw milk should be boiled (or pasteurized) before consumption. Sanitary measures should be taken at all stages of raw milk supply, from production to consumption, to provide dairy products to consumers. Awareness programs about potential hazards and antibiotic resistance in food of animal origin should be implemented. Monitoring the use of antibiotics in animals and humans is necessary to minimize the development of antibiotic resistance. Medical data on raw milk-associated illnesses should be collected in Pakistan. Emphasis should be placed on the zoonotic potential of *Escherichia coli* STEC and the role of dairy cattle in the spread of the pathogen. The Veterinary Public Health Department needs to regulate the production and sale of milk and milk products by introducing periodic screening and issuance of fitness certificates to farmers. Monitoring the use of antibiotics in animal and human therapy is necessary to minimize the development of antibiotic resistance in bacterial pathogens.

## Figures and Tables

**Figure 1 tropicalmed-09-00064-f001:**
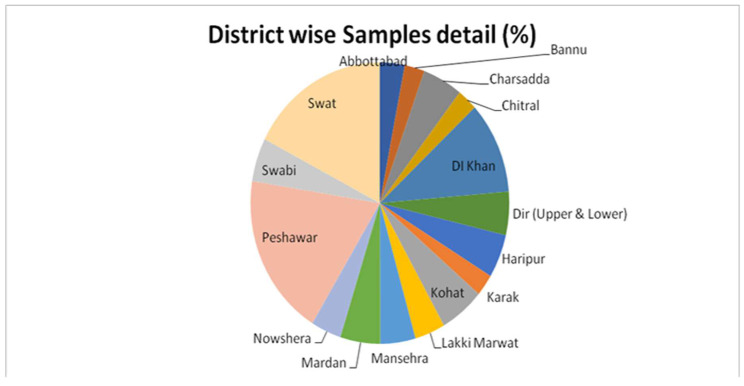
Prevalence of sample source (*n* = 800).

**Figure 2 tropicalmed-09-00064-f002:**
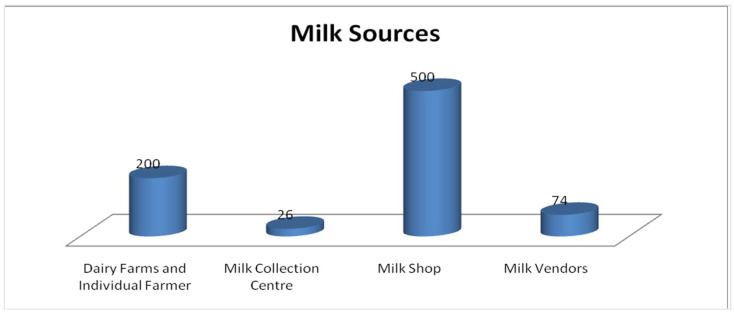
Distribution of raw milk samples collected from various sources.

**Figure 3 tropicalmed-09-00064-f003:**
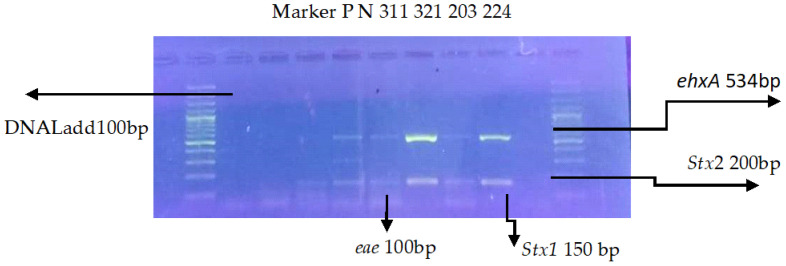
Multiplex PCR results for the presence of virulence genes (*eae* 100 bp, *Stx1* 150 bp, *Stx2*, 200 bp, *ehxA* 534 bp) in Shiga toxin-producing STEC. Lane 1 shows the 100 bp ladder, Lane 2 is the positive control, Lane 3 is the negative control, and Lanes 4–9 represent the test samples.

**Figure 4 tropicalmed-09-00064-f004:**
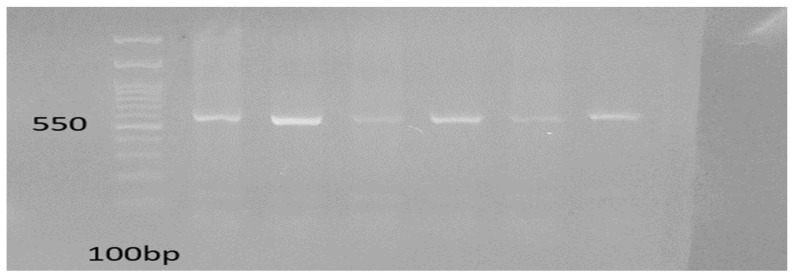
*bla_CTX-M_* gene amplified by PCR, marker 100 bp.

**Table 1 tropicalmed-09-00064-t001:** Primers, concentration, and annealing temperature used for molecular characterization of *E.coli,* and ESBL isolates. (Applied Biosystems, Waltham, MA, USA).

Genes	PCR Conditions	PCR Reaction Volume
*Stx1*, *Stx2*, *eae*, *ehxA*	1cycle	2.5 μL of 10× PCR buffer
	96 °C-10 min	0.15 mM MgCl_2_
	35 cycles	0.1 mM of each dNTP
	95 °C-45 s	0.5 μL of each primer
	60 °C-45 s	One unit of Taq DNA polymerase
	72 °C-45 s	3 μL of DNA
	1 Cycle final volume of 25 μL with sterile water 72 °C-8 min	
*bla_CTX-M_*, *bla_TEM_*, *bla_SHV_*	1cycle 2.5 μL of	2.5 μL of 10× PCR buffer 10× PCR
	96 °C-5 min	0.15 mM MgCl_2_ buffer
	25 cycles	0.1 mM of each dNTP (Thermoscientific, Waltham, MA, USA)
	95 °C-1 min	1.0 μLmM of *bla_CTX-M, TEM, SHV_* primers
	56 °C, 58 °C-1 min	One unit of Taq DNA polymerase
	72 °C-1 min	5 μL of DNA
	1 Cycle, 72 °C-10 min held 4 °C forever the final volume of 20 μL with sterile water	

**Table 2 tropicalmed-09-00064-t002:** Region-wise positive cases and probability statistics (*n* = 800).

Districts	*E. coli*		*E. coli* non-O157:H7		Total
	Positive	Not Detected	Positive	Not Detected	
Abbottabad	07	18	0	25	25
Bannu	10	10	1	19	20
Charsadda	21	19	2	38	40
Chitral	9	11	0	20	20
Dera Ismail Khan	32	53	6	79	85
Dir (Upper and Lower)	15	25	0	40	40
Haripur	12	28	0	40	40
Karak	8	12	0	20	20
Kohat	21	24	4	41	45
Lakki Marwat	12	18	1	29	30
Mansehra	7	28	0	35	35
Mardan	11	29	2	38	40
Nowshera	12	18	2	28	30
Peshawar	78	72	20	130	150
Swabi	19	21	2	38	40
Swat	47	93	0	140	140
Total	321	479	40	760	800

**Table 3 tropicalmed-09-00064-t003:** Distribution of raw milk samples collected from various sources positive for (*E. coli*, STEC, and ESBL).

Source Detail	*E. coli*		STEC		ESBL		Total
	Positive	Not Detected	Positive	Not Detected	Positive	Not Detected	
Dairy Farms and Individual Farmers	073	127	08	192	02	198	200
Milk Collection Centre	013	013	01	025	00	026	026
Milk Shops	207	293	30	470	08	492	500
Milk Vendors	028	046	01	073	01	073	074
Total	321	479	40	760	11	789	800
Probability Statistics	*p*-Value = 0.449		*p*-Value = 0.305		*p*-Value = 0.860		

**Table 4 tropicalmed-09-00064-t004:** Prevalence of *E. coli* (STEC) virulence genes (*Stx1*, *Stx2*, *eae*, *ehxA*) from bovine raw milk.

STEC Virulence Genes	*N* (%)
*Stx1*, *eae*	4 (10.0)
*Stx1*, *stx2*, *ehxA*	5 12.5)
*eae*	5 (12.5)
*Stx1*, *ehxA*	6 (15.0)
*ehxA*	4 (10.0)
*Stx1*, *eae*, *ehxA*	3 (7.5)
*Stx2*, *eae*, *ehxA*	2 (5.0)
*Stx1*, *stx2*	6 (15.0)
*Stx2*	3(7.5)
*Stx1*	2 (5.0)
Total	40 (100.0%)

**Table 5 tropicalmed-09-00064-t005:** Antimicrobial susceptibility testing of Shiga toxin-producing *E. coli* (STEC) isolated from raw milk in Khyber Pakhtunkhwa (CLSI 2020) (*n* = 22).

Antibiotic	Disc Concert	Resistance	%	Intermediate	%	Sensitive	%
Penicillin	P 10 IU	11	50	7	31	4	18
Amoxicillin	AML 30 µG	17	77	5	22	0	0
Amoxicillin/Clavulanic acid	AUG 30 µG	13	59	4	18	5	22
Sulphamathoxole	SMX 50 µG	9	40	8	36	5	22
Gentamicin	CN 10 µG	7	31	7	31	8	36
Streptomycin	S 10 µG	11	50	6	27	5	22
Oxytetracyline	OT 30 µG	6	27	8	36	8	36
Ceftriaxone	CRO 30 µG	12	54	6	27	4	18
Norfloxacin	NOR 10 µG	5	22	4	18	13	59
Enrofloxacin	ENR 5 µG	4	18	6	27	12	54
Florefencial	FFC 30 µG	6	27	5	22	11	50
Cefotaxime/Clavulnic acid	CTL 40 µG	8	36	9	40	5	22

Chi-Square = 37.01, *p*-Value = 0.023, RXC method using EpI-info 7.2 software.

## Data Availability

All the relevant data are provided in the article.

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
