# Peer review of "Multiple-Drug Resistant Shiga Toxin-Producing E. coli in Raw Milk of Dairy Bovine"

_tropicalmed, 2024, doi:10.3390/tropicalmed9030064_

Round 1

Reviewer 1 Report (Previous Reviewer 1)

Comments and Suggestions for Authors

General comments

The manuscript is much improved over the first version. In its present form, the manuscript can be read and understood by the reader, although some aspects of its formatting can still be greatly improved.

With respect to the scientific content, its degree of novelty is rather low and it is only a slight extension of current knowledge. However, due to the remoteness of the area of origin of the sampling, and the limited data available in general from this geographical area, it may have the interest to be published that it would not have had if it had been carried out in a more usual geographical area.

Specific comments:

Abstract section. In lines 29 and 34 is it referred to results obtained from E. coli. These results are referred to E. coli or to STEC?

Line 51. Improper use of capital letters. Both Hemorrhagic colitis or Haemolytic Urinary Syndrome are not proper nouns.

Line 106: Please insert an space between “10” and “ml”, in the way written for “90 ml” in the same line. Be consistent with formats throughout the manuscript.

Line 107 and throughout the manuscript: “hour” is most abbreviated as “h” in most articles, not as “hrs”.

Lines 120 and 125. There are two subheadings “2.3”. (Biochemical Characterization and Serotyping of the E. coli isolates”

Lines 131-135. Any reference strain was used as quality control?

Line 140 and throughout the manuscript: Culture media, reagents and instrumentation, should be cited included the name, city of manufacturing and country. In example. BHI could be “BHI (Difco, Detroit, USA)”. Please apply for all manuscript.

Line 167. Both Table 1 and table heading (and all other tables) were not formatted according to MDPI instruction for authors.

Line 182. Any reference strain was used as quality control?

Figure 1 is of low quality and seems to be deformed.

Figure 2 has not content sufficient to be included as a figure rather than in the text

Line 415. Please change “Conclusions and Recommendations” to “Conclusions”.

Line 421-422. “This study….Pakhtunkhwa” is materials and methods, not a conclusion.

Author contributions were not formatted according to MDPI´s guidelines for authors.

References, although improved, are still not formatted according to MDPI standards. For example, scientific names of bacteria are not in italics. The journal names are not in a homogeneous format with regard to the use of capital letters, etc. Some references (e.g., reference number 10, are incomplete). These types of issues should be very careful, since their omission gives the reader an image of a not very careful and rigorous work.

Author Response

Thank you for sending reviewers’ comments on our manuscript on,

Multiple-Drug Resistant Shiga Toxin-Producing E. coli in Raw Milk of Dairy Bovine in Khyber Pakhtunkhwa, Pakistan

We have revised our manuscript according to these comments. Answers to each of the comments are given below. I hope that this revised manuscript is now found suitable for publication,

Reviewer 1 Comments highlighted in a (Red color)

  1. Abstract section. In lines 29 and 34 is it referred to results obtained from E. coli. These results are referred to E. coli or to STEC?

Answer.  The results are attributed to STEC, which represents the pathogenic strains of E. coli. Both have been referenced.

  1. Line 51. Improper use of capital letters. Both Hemorrhagic colitis and Haemolytic Urinary Syndrome are not proper nouns.

Answer.  Changes have been done as per suggestion.

  1. Line 106: Please insert an space between “10” and “ml”, in the way written for “90 ml” in the same line. Be consistent with formats throughout the manuscript.

Answer.   As per suggestion changes has been done throughout the manuscripts

  1. Line 107 and throughout the manuscript: “hour” is most abbreviated as “h” in most articles, not as “hrs”.

Answer.   Abbreviation has been changed throughout the manuscript.

  1. Lines 120 and 125. There are two subheadings “2.3”. (Biochemical Characterization and Serotyping of the E. coli isolates”

Answer.    The sub heading has been changed and made in a proper sequence.

  1. Lines 131-135. Any reference strain was used as quality control?

Answer.   The suggestion has been implemented and a reference strain was utilized for quality      control.

  1. Line 140 and throughout the manuscript: Culture media, reagents and instrumentation, should be cited included the name, city of manufacturing and country. In example. BHI could be “BHI (Difco, Detroit, USA)”. Please apply for all manuscript.

Answer.    Changes has been  made and followed the suggestion.

  1. Line 167. Both Table 1 and table heading (and all other tables) were not formatted according to MDPI instruction for authors.

Answer.   All Tables headings and formats have been changed as per suggestions.

  1. Line 182. Any reference strain was used as quality control?

 Answer.    Reference Strain has been used

  1. Figure 1 is of low quality and seems to be deformed.

Answer.    Figure 1 size and quality has been upgraded

  1. Figure 2 has not content sufficient to be included as a figure rather than in the text

 Answer.     Figure 2 has elaborated the samples sources visibility and authenticity.

  1. Line 415. Please change “Conclusions and Recommendations” to “Conclusions”.

Answer.    As per suggestion changes has been made.

  1. Line 421-422. “This study….Pakhtunkhwa” is materials and methods, not a conclusion.

Answer.    Highlight the study area in the conclusion part.

  1. Author contributions were not formatted according to MDPI´s guidelines for authors.

Answer.    Changes has been made according to suggestion

  1. References, although improved, are still not formatted according to MDPI standards. For example, scientific names of bacteria are not in italics. The journal names are not in a homogeneous format with regard to the use of capital letters, etc. Some references (e.g., reference number 10, are incomplete). These types of issues should be very careful, since their omission gives the reader an image of a not very careful and rigorous work.

Answer.    The references format has been changed, bacteria are in italics, and reference 10 number were completed

Reviewer 2 Report (Previous Reviewer 2)

Comments and Suggestions for Authors

I rereviewed the study of Shiga toxin-producing E. coli (STEC non-O157:H7) in unpasteurized milk after the authors revised the article. Although the authors have improved in some aspects, there are still some deficiencies.

1. Incorrect writing format: Line 49,  "Food-borne" incorrectly uses italics, it is suggested to change it to" Food-borne".

2. Incorrect writing format: Line 151, "bla CTXM, bla SHV and bla TEM" format is not consistent with Line 319, Line 388" bla CTXM" format, it is recommended to use "bla CTXM". This problem occurs in many places in the article, and it is suggested to check the full text.

3. Incorrect writing format: Line 153 , "MgCl2" should be changed to" MgCl2".

4. Incorrect writing format: Line 252, the format of "TABLE 3" is not consistent with that of other table titles, and it is suggested to be modified.

5. Symbol use repetition: Line 323,  "22.5%.."

In summary, although the author has made some improvements to the article in the revised draft, there are still many problems of incorrect writing format, and it is suggested that the author should check the writing format more carefully in the future work. At the same time, in order to improve the readability and academic level of the article, it is suggested that the author further polish the language of the article and further improve the language quality of the article.

Comments on the Quality of English Language

Some format errors should be revised further.

Author Response

Thank you for sending reviewers’ comments on our manuscript on,

Multiple-Drug Resistant Shiga Toxin-Producing E. coli in Raw Milk of Dairy Bovine in Khyber Pakhtunkhwa, Pakistan

We have revised our manuscript according to these comments. Answers to each of the comments are given below. I hope that this revised manuscript is now found suitable for publication,

Reviewer 2 Comments highlighted in a (Blue color)

I rereviewed the study of Shiga toxin-producing E. coli (STEC non-O157:H7) in unpasteurized milk after the authors revised the article. Although the authors have improved in some aspects, there are still some deficiencies.

  1. Incorrect writing format: Line 49,  "Food-borne" incorrectly uses italics, it is suggested to change it to" Food-borne".

Answer. Changes has been made

  1. Incorrect writing format: Line 151, "bla CTXM, bla SHV and bla TEM" format is not consistent with Line 319, Line 388" bla CTXM" format, it is recommended to use "bla CTXM". This problem occurs in many places in the article, and it is suggested to check the full text.

Answer. Changes has been made throughout the manuscript

  1. Incorrect writing format: Line 153 , "MgCl2" should be changed to" MgCl2".

Answer. Changes has been made

  1. Incorrect writing format: Line 252, the format of "TABLE 3" is not consistent with that of other table titles, and it is suggested to be modified.

Answer. Table formats has been changed and modified

  1. Symbol use repetition: Line 323,  "22.5%.."

Answer. Repetition has been deleted

In summary, although the author has made some improvements to the article in the revised draft, there are still many problems of incorrect writing format, and it is suggested that the author should check the writing format more carefully in the future work. At the same time, in order to improve the readability and academic level of the article, it is suggested that the author further polish the language of the article and further improve the language quality of the article.

Answer. The quality of English in the manuscript has been enhanced to meet the requirements of the journal and ensuring that the manuscript adheres to the language standards specified by the journal.

Round 2

Reviewer 1 Report (Previous Reviewer 1)

Comments and Suggestions for Authors

The authors have corrected most of the problems mentioned in the previous version, but formatting issues regarding tables, figures and references are still present. Perhaps such formatting discrepancies can be fixed during editing of the text, so no further revision is required.

This manuscript is a resubmission of an earlier submission. The following is a list of the peer review reports and author responses from that submission.

Round 1

Reviewer 1 Report

Comments and Suggestions for Authors

The work is written in a tremendously sloppy manner, it gives the impression of being a draft and not a final version. The authors have not followed the instructions for authors of the journal or the publisher. Even though they have not followed these instructions, they have not been uniform in the format presented, since in some cases the author's last name and year have been cited, in other cases the full name (line 133). The SI has not been followed in many of the units described (for example, time is abbreviated as hr). The methods are basic for today's times and in some cases totally inadequate. For example, an inappropriate reference is used in the disk diffusion, the CLSI document is not cited, and a collection strain has not been used as an internal control. It is mentioned that in the statistical analysis the t-test has been used to "evaluate any important association among the antibiotics resistance frequencies". Student's t-test is used to compare means, not to determine associations. As a clear example of extreme disorganization, the first table to appear in the text is number 5, and the second (Table 4) appears in the section of materials and methods, when in fact it is a result.

These are only examples, since both the introduction and the discussion are rather a compilation of single sentences without any link between them, and in many cases, repeating information that had already been previously cited.

Comments on the Quality of English Language

Both the introduction and discussion sections are rather a compilation of single sentences without any link between them, and in many cases, repeating information that had already been previously cited. There are a lot of grammar mistakes, at a very basic level.

Author Response

Author's reply attached

Reviewer 2 Report

Comments and Suggestions for Authors

This research paper investigates the presence of Multiple-Drug Resistant Shiga Toxin-Producing E. coli in Raw Milk of Dairy Bovine in Khyber Pakhtunkhwa, Pakistan. I have reviewed this paper and would like to provide the following comments:

1.       The data collection method is not clearly specified: Line 100, The study does not provide detailed information on the selection criteria and collection methods for raw milk samples, which may affect the reliability and consistency of the data.

2.       The writing format is inconsistent: Line 113, The temperature unit "37℃" in 2.2 is not consistent with the "℃" style used later in the article.

3.       The table formatting sequence in the article is incorrect: Line 144 and 146, Table 4 and Table 5 are placed before Table 1, which makes it inconvenient for readers to navigate.

4.       The writing format is inconsistent: Line 146 and 147, "Safir Table 5. prevalence of E. coli (STEC) virulence genes (stx1, stx2, eae and ehxa) from bovine raw 146 milk (n = 800) different areas of Khyber Pakhtunkhwa n=40." The initial letter is not capitalized, the sample size "n" is inconsistent with the "N" in other tables, and it is recommended to check the titles of all tables and figures.

5.       The writing format is inconsistent: Line 168 to 176, The formatting of the numerical units in 2.7 is inconsistent, with "Ml" and "ml", "mM", "sc", and "seconds". It is recommended to standardize the format.

6.       The writing format is inconsistent: Line 180 and Line 221 , 222, 229, The formatting of "bla Ctxm" and "bla Tem" is inconsistent with the writing format used later in the article. It is recommended to unify the writing format.

7.       The "Conclusions and Recommendations" section is too concise and lacks a comprehensive interpretation of the research results: It is suggested to further explore the significance of the findings and provide specific practical recommendations for addressing multidrug-resistant E. coli in raw milk. Additionally, the "Recommendations" section at the end can mention future research directions.

Comments on the Quality of English Language

The language must be improved to fit the requirement of the journal as well as easy to understand by the readers. 

Author Response

Author's reply attached

Reviewer 3 Report

Comments and Suggestions for Authors

The study describes in this manuscript aims to determine the prevalence of MDR Shiga Toxin E. coli (STEC) in raw milk from dairy farms, street vendors and milk shops from different parts of Pakistan. The topic and the objectives of the study were relevant since STEC is a food-borne pathogen that can be transmitted to the consumers via raw milk. However, calculating a prevalence in a sampling collection from various origins creates a bias in prevalence results since a same strain can be isolated from dairy farm and in milk shop then after, unless the authors have ensured that there are no duplicates in their samples (not stated in the document). The bacterial isolation step describes in materials and methods section seems to be a selective isolation since the broth used for enrichment is supplemented with ceftrixime. What is this molecule? An antibiotic (not known)? In any case, the isolated strains seem to have been selected on the basis of a particular phenotype, which means that the results presented may not be linked to the E. coli population, but to a specific E. coli population (resistant to an antibiotic family?). The material and method are very confusing. For instance, it is not vogues prosker test but Voges-Proskauer test. Suppliers are not specified for the majority of consummables. Too much details sometimes (&2.7; why not refer to the table 3?), not enough otherwise (&2.5 ; why refer to Patel et al, line152? this author is not included in the references list).  Could the authors explain why there are p-values in table 1 and in table 2 (which describe positive and negative samples for E. coli or STEC) as in &2.8, the authors described that the t-test was used to explain the antibiotic resistance frequencies?

Throughout the manuscript, naming of scientific terms as antibiotics, genes, bacteria, system of units are not always consistent with the international nomenclature and should be revised. Many typos throughout the document (24hr instead of 24 h; capital letter in the middle of a sentence; 2 verbs in the same sentence, °C to be revised...).

All these problems of formatting, writing, lack of precision and rigour, don't allow the reviewer to properly assess the quality of the methods used in this study, the quality of the results and their relevance to the subject. 

For all the weakness describe above, this paper must be revised before reconsidering a submission.

Author Response

Author's reply attached
